



# The role of Rossby waves in polar weather and climate

Tim Woollings[1], Camille Li[2], Marie Drouard[1,3], Etienne Dunn-Sigouin[4], Karim A. Elmestekawy[1], Momme Hell[5], Brian Hoskins[6], Cheikh Mbengue[1,7], Matthew Patterson[1], and Thomas Spengler[2]

[1]Atmospheric, Oceanic and Planetary Physics, Parks Rd, Oxford, OX1 3PU, UK
[2]Geophysical Institute, University of Bergen, and Bjerknes Centre for Climate Research, Bergen, Norway
[3]Earth Physics and Astrophysics Department, Universidad Complutense de Madrid, Spain
[4]NORCE Norwegian Research Centre AS and Bjerknes Centre for Climate Research, Bergen Norway
[5]Department of Earth, Environmental, and Planetary Sciences, Brown University, Providence, RI, USA
[6]Grantham Institute, Imperial College London, London, UK, and Department of Meteorology, University of Reading, Reading, UK
[7]Climate Modeling Alliance, California Institute of Technology, Pasadena, California, USA

**Correspondence:** Tim Woollings (tim.woollings@physics.ox.ac.uk)

**Abstract.** Recent Arctic warming has fuelled interest in the weather and climate of the polar regions and how this interacts with lower latitudes. Several interesting theories of polar-midlatitude linkages involves Rossby wave propagation as a key process even though the meridional gradient in planetary vorticity, crucial for these waves, is weak at high latitudes. Here we review some basic theory and suggest that Rossby waves can indeed explain some features of polar variability, especially when
relative vorticity gradients are present.

We suggest that large-scale polar flow can be conceptualised as a mix of geostrophic turbulence and Rossby wave propagation, as in the mid-latitudes, but with the balance tipped further in favour of turbulent flow. Hence, isolated vortices often dominate but some wavelike features remain. As an example, quasi-stationary or weakly westward-propagating subpolar anomalies emerge from statistical analysis of observed data, and these are consistent with some role for wave propagation.
The noted persistence of polar cyclones and anticyclones is attributed in part to the weakened effects of wave dispersion, the mechanism responsible for the decay of mid-latitude anomalies in downstream development. We also suggest that the vortex-dominated nature of polar dynamics encourages the emergence of annular mode structures in principal component analyses of extratropical circulation.

Finally, we consider how Rossby waves may be triggered from high latitudes. The linear mechanisms known to balance
localised heating at lower latitudes are shown to be less efficient in the polar regions. Instead, we suggest the direct response to sea ice loss often manifests as a heat low, with radiative cooling balancing the heating. If the relative vorticity gradient is favourable this does have the potential to trigger a Rossby wave response, although this will often be weak compared to waves forced from lower latitudes.



## 1 Introduction

Rapid Arctic warming over recent decades has spurred widespread interest in the weather and climate of the polar regions (Screen et al., 2018a; Overland et al., 2015). Historically viewed as a quiet, largely anticyclonic zone, the Arctic is now known to host a range of weather systems including vigorous cyclone activity (Walsh et al., 2018). Polar atmospheric circulation has been found to play an important role in driving anomalies in sea ice cover, with individual Arctic cyclones (Simmonds and Rudeva, 2012), anticyclones (Wernli and Papritz, 2018) and large scale circulation patterns (Overland and Wang, 2010) all implicated in rapid declines in ice extent. The polar regions are also key centres of action of the leading patterns of atmospheric variability in both hemispheres, the annular modes (Thompson and Wallace, 2000; Spensberger et al., 2020).

The potential influence of the Arctic on mid-latitude climate has been a prominent, and often public, focus of climate science (Cohen et al., 2014; Barnes and Screen, 2015; Wallace et al., 2014). The ability of Arctic warming to influence the mid-latitude jet speed and position seems clear (Screen et al., 2018b; Baker et al., 2017), particularly on its polar flank (Harvey et al., 2015). Proposals of Arctic influence on mid-latitude waves and circulation variability (Francis and Vavrus, 2012; Petoukhov et al., 2013) have received limited support from theory (Hoskins and Woollings, 2015), models (Cattiaux et al., 2016), and some observational analyses (Blackport and Screen, 2020). Despite this, considerable interest remains in the coupling between polar and mid-latitude dynamics, which is clearly important for understanding and predicting both regions on both weather and climate timescales (Jung et al., 2014; Barnes and Simpson, 2017; Zappa et al., 2018).

Atmospheric Rossby waves, comprising alternating cyclonic and anticyclonic vorticity anomalies, are dominant structures of both mean climate and variability in the mid-latitudes (Rossby, 1939; Blackmon, 1976). Rossby waves owe their existence to the meridional gradient in absolute vorticity, which imposes strong constraints on circulation in both the tropics and mid-latitudes. A key component of this is due to the meridional variation of the planetary vorticity, commonly measured by the meridional gradient of the Coriolis parameter and referred to as the 'beta-effect'. Although the Coriolis parameter itself is large in the polar regions, its meridional gradient is weak and approaches zero at the poles. Hence, the beta-effect is weak and so the conditions are generally less favourable for Rossby wave propagation than they are at lower latitudes. Despite this, Rossby wave mechanisms are often invoked in studies of polar climate and its lower-latitude linkages, for example:

1. Both stationary and transient Rossby waves, perhaps triggered in the tropics, have been implicated in the transports of heat and moisture into the Arctic which have contributed to recent warming (Ding et al., 2014; Lee, 2014; Graversen and Burtu, 2016; Dunn-Sigouin et al., 2021).

2. Amplified and/or persistent waves at mid to high latitudes have been claimed to be linked to Arctic warming, with suggestions of causality in both directions (Francis and Vavrus, 2012; Woods et al., 2013; Mann et al., 2017; Kornhuber et al., 2017).

3. The recent reduction in Barents-Kara sea ice has been proposed to have triggered a stationary Rossby wave pattern which contributed to a relative cooling trend in recent Eurasian winters (Honda et al., 2009; Kretschmer et al., 2016; Mori et al., 2019; Cohen et al., 2019), potentially via an impact on the stratosphere (Kim et al., 2014).



Based on this literature, Rossby waves seem surprisingly important for polar dynamics, despite the weakness of the beta-effect. The aim of this paper is therefore to review some established aspects of tropospheric Rossby wave dynamics and apply
these to a discussion of the role of Rossby waves in the polar regions, loosely defined here at the regions poleward of 65° of latitude. This will hopefully facilitate the further application of dynamical concepts to the understanding of the changing weather and climate of the polar regions.

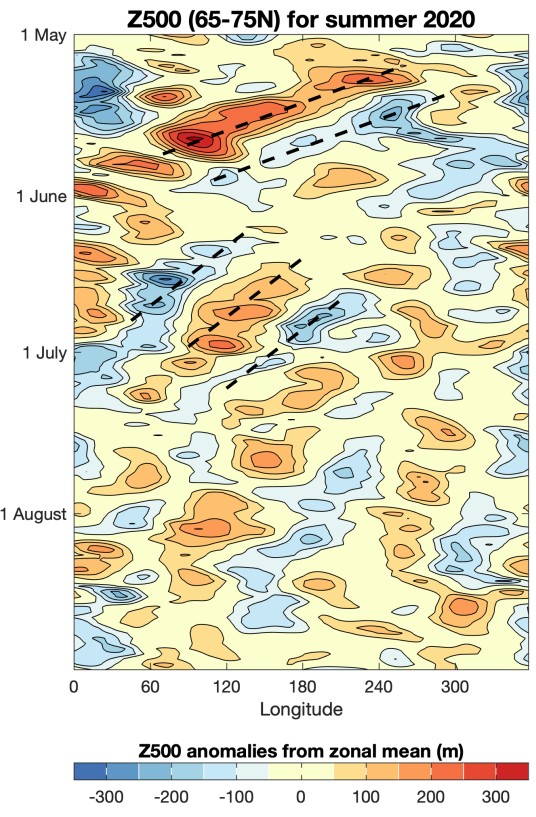

**Figure 1.** Hovmöller plot of 500hPa geopotential height from the summer of 2020, using daily data from the NCEP reanalysis. Dashed lines are drawn to indicate suggested examples of phase propagation.

A preview of this discussion is given by some examples from the Arctic circulation in 2020. Figure 1 shows a Hovmöller
diagram of the 500mb geopotential height to provide an overview of the circulation. For May and June, the circulation is
dominated by large-scale wavelike structures with alternating cyclonic and anticyclonic features. These show characteristics of
Rossby wave evolution which are similar to the familiar mid-latitude structures (Wirth et al., 2018), in particular in having a
distinct phase velocity of individual centres and group velocity, by which the wave energy is fluxed into new centres of action to
the east. Some notable differences are apparent, however: 1) the individual centres are more persistent, lasting over two weeks
in some cases, and 2) the phase velocity is directed to the west, in contrast to midlatitude cases where anomalies generally

propagate eastward with the mean flow. The resulting persistent, slowly-moving anomalies have been linked to anticyclonic
conditions and/or southerly flow which fuelled intense episodes of the Siberian heatwave and wildfire activity that occurred
through May and June (Overland and Wang, 2021). Later in the summer the circulation shows much less organised wavelike
structure, with cyclonic and anticyclonic vortices of different sizes and little obvious phase propagation. The prominent low
height anomaly near the date line marks the presence of an extreme and persistent Beaufort Sea cyclone towards the end of

July. As for other summer Arctic cyclones, the associated surface winds led to notable sea ice divergence[1]. Based on these
examples, we can anticipate that Rossby waves sometimes, but not always, play a role in polar dynamics, and that in these
cases their physical characteristics are likely different from mid-latitude waves.

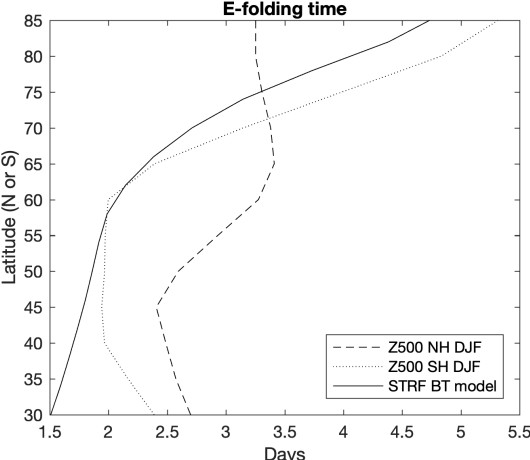

**Figure 2.** E-folding timescale as a function of latitude for Z500 observational data from the NCEP2 reanalysis and streamfunction from a
barotropic model simulation.

As further motivation, Figure 2 shows the dependence of the eddy decay timescale on latitude. This is derived from the lag
autocorrelation function of 500mb geopotential height, sampled every $30°$ of longitude. Daily NCEP2 reanalysis data from

DJF months from 1979 to 2018 was used, giving situations with strong stationary waves in northern hemisphere winter and
weak stationary waves in southern hemisphere summer. The decay timescale here is quantified by the e-folding timescale,
using linear interpolation to find the time lag at which the autocorrelation drops to $1/e$. In both cases, the mid-latitude regions
exhibit short timescales of 2 to 2.5 days, consistent with the frequent passage of synoptic systems / baroclinic wave packets
in the storm tracks. The timescale lengthens, however, as the latitude increases, by an extra day in the northern case and up to

three days in the southern case. These longer timescales are consistent with the impression given by the examples in Figure 1.
For comparison, Figure 2 also shows a similar diagnostic derived from the streamfunction of a barotropic model simulation,
as used by Patterson et al. (2020). Following Vallis et al. (2004), the barotropic vorticity equation is solved on the sphere and
forced by stirring in the mid-latitudes to represent the effects of baroclinic growth within the storm track (see the Appendix

---

[1]https://nsidc.org/arcticseaicenews/2020/08/



for more details). The comparison of this highly idealised system to observations is purely qualitative, however the similar

behaviour of the decay timescales suggests that barotropic Rossby wave dynamics may play a fundamental role in setting the

latitudinal variations in timescale in the observations.

In the rest of this paper we discuss how theoretical predictions of Rossby wave behaviour can, at least qualitatively, explain

observed characteristics such as the longer persistence times at high latitudes. In Section 2 we revise some of the relevant

theories on Rossby waves, with a focus on polar latitudes. Following this, we present some observational analyses of the

potential for wave propagation (Section 3), the balance between wavelike and turbulent flow (Section 4), and the dependence

of flow structures on latitude (Section 5). Finally, in section 6 we investigate the mechanisms by which Rossby waves might

be triggered in the polar regions.

## 2 Rossby wave theory

### 2.1 The beta plane

The variation of the Coriolis parameter $f = 2\Omega \sin \phi$ with latitude $\phi$ near a fixed latitude $\phi_0$ can be approximated as a Taylor

series

$$
\begin{aligned}
f &\approx 2\Omega \left( \sin \phi_0 + (\phi - \phi_0) \cos \phi_0 - \frac{(\phi - \phi_0)^2}{2} \sin \phi_0 + ... \right) \\
&\approx f_0 + \beta(y - y_0) - \gamma(y - y_0)^2,
\end{aligned}
\tag{1}
$$

$$
\tag{2}
$$

where $a$ and $\Omega$ are the radius and angular velocity of Earth, respectively. The cartesian coordinate $y = a\phi$ is used and the

parameters are $\beta = \frac{2\Omega}{a} \cos \phi_0$ and $\gamma = \frac{\Omega}{a^2} \sin \phi_0$ (though note that $\delta$ is sometimes used in place of $\gamma$). The Taylor series becomes

divergent at very high latitudes, although this is avoided in a similar form derived in a rotated coordinate system by Harlander

(2005). The commonly used $\beta$-plane approximation neglects the $\gamma$ term and hence $f$ is assumed to vary linearly with meridional

gradient $\beta$ on a tangent plane to the sphere. This linear variation suffices to allow Rossby waves to develop; for example an air

mass moving poleward will experience an increase in planetary vorticity $f$ according to the $\beta$ term, and will therefore acquire

negative, or anticyclonic relative vorticity to compensate. $\beta$ decreases towards zero with latitude, such that at $70°$ of latitude $\beta$

is roughly half as large as it is at $45°$ (Figure 3).

### 2.2 Barotropic vorticity equation

The simplest model capable of supporting Rossby waves is the barotropic vorticity equation on a $\beta$-plane:

$$
\left( \frac{\partial}{\partial t} + u \frac{\partial}{\partial x} + v \frac{\partial}{\partial y} \right) \zeta + \beta^* v = 0,
\tag{3}
$$

where $u$ and $v$ are the velocities in the cartesian coordinates $x$ and $y$ on the local tangent plane to the sphere, and $\zeta$ is the

relative vorticity. The gradient in absolute vorticity is given by $\beta^* = \beta - \frac{\partial^2 u}{\partial y^2}$, where the relative vorticity gradient is included



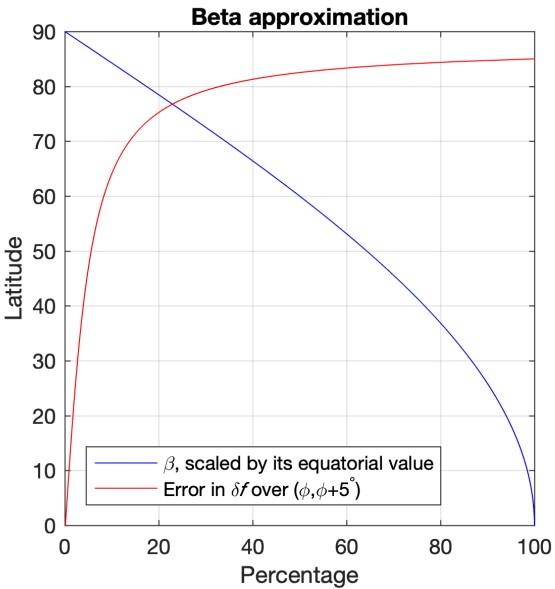

**Figure 3.** Variation with latitude of $\beta$ and also the error in $\delta f$ over a distance of $5°$ poleward of a given latitude when using the $\beta$-plane approximation.

alongside the beta-effect. To identify wave solutions, we linearise about a uniform background zonal flow $u = U + u'$ and search for solutions of the form $\exp i(kx + ly - \omega t)$, where $k$ and $l$ are the zonal and meridional wavenumbers and $\omega$ is the frequency (see e.g. Vallis, 2006). Under this linearisation $\beta^*$ reverts to $\beta$ and we obtain the well known dispersion relation for Rossby waves

$$\omega = Uk - \frac{\beta k}{K^2}, \tag{4}$$

where $K$ is termed the total wavenumber and is given by $K^2 = k^2 + l^2$. Hence, if $\beta$ was zero then the only time dependence would be due to steady advection by the background flow. With non-zero $\beta$, however, Rossby waves can propagate with zonal phase velocity

$$c_x = \frac{\omega}{k} = U - \frac{\beta}{K^2}. \tag{5}$$

Waves are therefore propagating upstream against the background flow, and this upstream propagation is faster for long waves with smaller $k$. Hence, long waves have the potential for westward phase propagation over the ground, while short waves will instead progress eastward. The group velocity is

$$\mathbf{c_g} = \left( \frac{\partial \omega}{\partial k}, \frac{\partial \omega}{\partial l} \right) = (c_x, 0) + \frac{2\beta}{K^2} \cos(\alpha) \, \hat{\mathbf{K}}, \tag{6}$$

where $\alpha$ is the angle between the x-axis and the wavevector $\mathbf{K} = (k, l)$. The zonal group velocity is therefore greater than the zonal phase velocity. For realistic parameters, the zonal group velocity is usually directed eastward, although there are




exceptions when $k$ is small and $l$ is large (O'Reilly et al., 2018). The dependence of the phase velocity on wavelength shows that the wave is dispersive, with the envelope of a wave packet propagating more quickly eastward than the individual ridges
and troughs of the Rossby wave. This is the fundamental mechanism of downstream development, by which new weather systems form ahead, i.e. to the east, of existing ones (Simmons and Hoskins, 1979; Chang, 1993).

Based on this theory, we are already in a position to make some predictions on how Rossby waves might behave at high latitudes where $\beta$ is small. Firstly, and somewhat trivially, the propagation of Rossby waves will be weaker, for example with a weaker upstream phase velocity against the background flow. Secondly, equation 5 shows that the dependence of phase velocity
on wavenumber will be weaker if $\beta$ is smaller, so that the waves will be less dispersive. This is also seen in equation 6, which shows that the zonal phase and group velocities differ by a quantity proportional to $\beta$. We therefore expect the phase and group velocities to approach each other as latitude increases, so that the process of downstream development should be inhibited.

### 2.3   Beta plane limitations

While $\beta$ decreases with latitude, $\gamma$ increases, so the error associated with the $\beta$-plane approximation also increases (equation 2).
The $\beta$-plane overestimates the latitudinal variation of Coriolis on the poleward side of the central latitude, and underestimates it on the equatorward side. As an example, consider the difference in $f$ between $\phi_0$ and a point five degrees of latitude further poleward. The $\beta$-plane overestimates this difference by 8% at 60° latitude, by 14% at 70° and by 33% at 80° (Figure 3). Hence, as concluded by Harlander (2005), $\gamma$ effects are increasingly important polewards of 60°. These errors in the latitudinal variation of Coriolis lead directly to errors in the phase speeds of Rossby waves, as seen from equation 5. A $\beta$-plane model will
therefore overestimate the intrinsic upstream phase speed $\beta/K^2$ in the poleward region of the tangent plane, and underestimate it in the equatorward region. These errors are again greater for tangent planes located at higher latitudes.

In some planetary atmosphere applications the $\gamma$ effect is important enough that a $\gamma$-plane framework is used, wherein the $\beta$ term might even be neglected but the $\gamma$ term retained. In Earth's atmosphere, however, $\gamma$ effects appear to lead to relatively minor changes in the behaviour of Rossby waves. For example, Yang (1987, 1988) found $\gamma$ effects to alter the orientation
of ridge/trough lines in theoretical waves, and also to increase the meridional extent of a wave packet by around 10-20%. This stretching is consistent with the weaker vorticity gradient at high latitudes, as the stronger gradient inhibits meridional wavelengths in the mid-latitudes (see section 4). Nof (1990) investigated the motion of isolated monopole vortices in the presence of both $\beta$ and $\gamma$ effects. Intriguingly, the $\gamma$ terms cancelled out in the migration rate, due to their symmetry. Hence, even though $\gamma$ is of the same order as $\beta$ in the polar region, an isolated vortex in this model will drift westward according to
the local $\beta$.

### 2.4   Wave activity considerations

The propagation of Rossby waves in the latitude-height plane is often characterised in the quasigeostrophic system by the Eliassen-Palm relation (Andrews and Mcintyre, 1976; Vallis, 2006)

$$\frac{\partial A}{\partial t} + \nabla \cdot \mathbf{F} = D. \tag{7}$$





This is a conservation law for the wave activity density $A = \frac{1}{2} \frac{\overline{q'^2}}{\partial \bar{q}/\partial y}$, where $q$ is the quasigeostrophic potential vorticity (PV), $\mathbf{F}$ is termed the Eliassen-Palm flux and $D$ represents forcing and dissipation. Overbars indicate a zonal average, and primes a deviation from this. Under the assumptions that the waves are small amplitude and slowly modulated, the flux can be written as $\mathbf{F} = \mathbf{c_g} A$, i.e. as the product of the group velocity and the wave activity density.

Based on this framework, we can make some brief comments on the latitude-dependence of Rossby waves. If we assume

that $D = 0$ then

$$\left( \frac{\partial}{\partial t} + \mathbf{c_g} \cdot \nabla \right) A + A \nabla \cdot \mathbf{c_g} = 0. \tag{8}$$

In the case that the group velocity is non-divergent, the wave activity density will then be constant following the direction of energy propagation of the wave. The PV gradient $\partial \bar{q}/\partial y$ is simply equal to $\beta$ for uniform flow on a beta-plane (or more generally $\beta^*$ in the presence of a mean relative vorticity gradient). Hence, if a wave propagates to a higher latitude where $\beta$ is

weaker, then the potential vorticity variance $\overline{q'^2}$ must also decrease in order to conserve $A$. However, in the small amplitude case we can approximate $q'$ by $\delta y \frac{\partial \bar{q}}{\partial y}$, where $\delta y$ represents a small meridional displacement of a fluid parcel. Then $A \approx \beta \overline{\delta y^2}/2$ so that, although the amplitude of PV perturbations will decrease as the wave moves to a higher latitude, the meridional displacement of air parcels will increase.

In more realistic situations wave activity is seen to converge and diverge at different latitudes. By combining equation 7 with

the zonal momentum equation we can obtain a version of the non-acceleration theorem (Vallis, 2006):

$$\frac{\partial}{\partial t} (\overline{u} + A) = 0. \tag{9}$$

This shows that a gain of wave activity at a particular latitude can only be achieved at the expense of the zonal mean wind at that latitude. A poleward propagating wave transporting wave activity to high latitudes will then weaken the zonal wind and hence the relative vorticity gradient and make the region even less favourable for wave propagation. As a result, high

latitude regions are prone to convergence of wave activity and hence, when zonal winds are weak, to frequent breaking of Rossby waves involving irreversible overturning of PV contours (McIntyre and Palmer, 1983; Gabriel and Peters, 2008). This process often results in persistent cut-off anticyclones, sometimes referred to as high-latitude blocks (Woollings et al., 2008), which can have severe impacts on the polar regions (Hanna et al., 2014; Madonna et al., 2020) as well as mid-latitudes (e.g. Franzke et al., 2004; Strong and Magnusdottir, 2008). The linear theory used above breaks down when the waves are no longer

slowly modulated and small amplitude, although finite amplitude counterparts to these theories have recently been developed to address these issues (Nakamura and Zhu, 2010; Methven and Berrisford, 2015; Nakamura and Huang, 2018). However, linear wave theory can provide further useful guidance on the behaviour of waves at high latitudes, in particular in the example of the ray tracing of stationary waves on the sphere outlined next.

## 2.5  Ray tracing theory

Hoskins and Karoly (1981) described the ray tracing approach for the linearised barotropic vorticity equation, as above, but using a Mercator projection to account for the spherical geometry. Meridional gradients in relative vorticity are included in





addition to the planetary gradient $\beta$. The resulting dispersion relation

$$\omega = \bar{u}_M k - \frac{\beta_M k}{K^2} \qquad (10)$$

is identical to equation 4, except that $U$ is replaced by $\bar{u}_M(y)$, the zonal velocity in a Mercator projection, and $\beta$ with $\beta_M$, the

meridional gradient of absolute vorticity on the sphere multiplied by the cosine of latitude.

The theory defines rays as being parallel to the local group velocity at which the wave energy propagates. Setting $\omega = 0$ to focus on stationary waves, ray tracing proceeds by fixing a zonal wavenumber $k$ of interest and a starting point for the ray. The meridional wavenumber $l$ is then determined from the dispersion relation, given local values of $\bar{u}_M$ and $\beta_M$. The local ray direction is then given by $(k, l)$ and an increment is taken in this direction before recalculating $l$ for the next step. Since

equation 10 is a quadratic expression for $l$ it gives both positive and negative solutions, relating to poleward and equatorward directed rays respectively. Here we will focus on the poleward moving rays associated with waves forced at low latitudes moving towards the polar regions.

The total stationary wavenumber is a key element of the theory. Setting $\omega = 0$ in equation 10, this is given by

$$K_s = \left( \frac{\beta_M}{\bar{u}_M} \right)^{1/2}, \qquad (11)$$

and is a function of latitude and longitude only, for a given basic flow. In practise, $K_s$ is often a decreasing function in latitude. Hence, $l$ must decrease as a ray progresses poleward with constant $k$, so that the ray path becomes more zonal. This continues to the point where $l = 0$ and the ray points directly east. Here, the wave has reached its turning latitude where its zonal wavenumber $k$ matches the local $K_s$ and the wave henceforth returns equatorward. The wave can be thought of as refracting back towards higher values of $K_s$ (Hoskins and Ambrizzi, 1993).

Ray tracing theory relies on the WKBJ approximation of a slowly-varying background flow relative to the length scale of the wave. This approximation has limitations, in particular in application to more realistic background flows, and ray tracing is often used only for qualitative understanding as a result (Potter et al., 2013; Wirth, 2020; Li et al., 2020). With this in mind, we note here a few implications for Rossby waves at high latitudes, following Hoskins and Karoly. Firstly, the decrease in $\beta$, and hence $K_s$ at high latitudes shows that waves propagating towards the polar regions will turn and refract back equatorwards.

This is anticipated to be a common situation, given the strong wave sources associated with deep convection in the tropics and with baroclinic instability in mid-latitudes.

Secondly, the theory predicts that longer waves should propagate to higher latitudes than shorter waves. This is because the turning latitude, at which waves refract equatorwards, is set by the latitude where $k = K_s$, and again $K_s$ generally decreases with latitude. As a result, Rossby wave activity at high latitudes is expected to be dominated by the lower zonal wavenumbers,

although due to the polar convergence of meridians the physical length scales of these waves are much shorter than at low latitudes.

Hoskins and Karoly (1981) also argued that the amplitude of streamfunction perturbations associated with a wave should increase as it propagates to higher latitudes, a feature that was also reproduced in their model experiments. The product $v_g A$ is constant following a ray, and the decrease of the meridional group velocity $v_g$ as the wave approaches the turning





latitude implies an increase in wave activity density. Specifically, $l\hat{\psi}^2$ is conserved along rays, where $\hat{\psi}$ is the magnitude of streamfunction perturbations. Conversely, this implies that waves initiated at a high latitude are predicted to weaken in amplitude as they propagate to lower latitudes where $l$ increases.

Finally, Karoly (1983) and Yang and Hoskins (1996) extended this approach to consider non-stationary waves, i.e. those with non-zero frequency and hence phase speed. While some of the qualitative aspects of wave propagation were unchanged,

some interesting differences were noted which are relevant to high latitudes. Wave propagation only occurs when $U - c_x > 0$; for a westerly flow this condition is always satisfied for disturbances moving intrinsically westward ($\omega, c_x < 0$) while for intrinsically eastward moving disturbances ($\omega, c_x > 0$) the intrinsic phase speed has to be weaker than the background flow. Hence, a critical line (where $U = c_x$) occurs which inhibits propagation for the low wavenumbers with rapid eastward phase propagation. As a consequence, we might expect westward-propagating waves of low wavenumbers to be more common at

high latitudes, although the frequency of the lower-latitude wave forcing will also be important in determining the phase speed. The examples of westward propagation in Figure 1 suggest that this theory may indeed be relevant.

The effect of westward phase propagation on group propagation can be inferred by rewriting the group velocity of equation 6 as

$$\mathbf{c_g} = \left( \frac{\partial \omega}{\partial k}, \frac{\partial \omega}{\partial l} \right) = c_x \mathbf{i} + 2(U - c_x) \cos(\alpha) \hat{\mathbf{K}}. \tag{12}$$

Hence, the direction of group propagation is no longer parallel to the wave vector $\hat{\mathbf{K}}$ but for $c_x < 0$ is directed more meridionally as the zonal component is reduced. This prediction was confirmed by Yang and Hoskins (1996) using model experiments and ray tracing with non-zero frequency, both of which showed wave paths approaching the high latitudes directly before being turned back at the turning latitude.

### 2.6 Summary

Based on this theoretical discussion, how might we expect Rossby waves at high latitudes to differ from those in mid latitudes? Firstly, we should expect wave propagation to be weaker in general, and associated with larger meridional parcel displacements due to the weaker vorticity gradient. Only the lowest zonal wavenumbers are expected to reach the polar regions, and by conservation of wave activity we expect these to show increased wave amplitude at high latitudes. Waves with a westward phase speed may be common, as they are not restricted by the presence of critical lines. In these cases, wave rays are expected

to follow more meridional pathways than expected from stationary wave theory. Finally, the similarity of phase and group velocities at high latitudes is expected to inhibit the process of downstream development which is a key mechanism of eddy decay at lower latitudes.

### 3 Potential vorticity gradients

The stationary wavenumber $K_s$ is a useful diagnostic of the ability of a given background flow to support Rossby waves. Maps

of the climatological $K_s$ are given by Hoskins and Ambrizzi (1993) and Hoskins and Woollings (2015), and these highlight the



importance of the midlatitude waveguides for Rossby wave propagation. However, Wirth (2020) noted the limitations of ray tracing theory and suggested that the potential vorticity (PV) gradient could be used as a reliable proxy for waveguidability. Both $K_s$ and the PV gradients include the effects of relative as well as planetary vorticity gradients, and so can reflect the potential for wave propagation even at high latitudes where $\beta$ is small. Here, we consider the climatological meridional gradients

of the potential temperature on the 2PVU potential vorticity surface, $\theta_{PV2}$, in Figure 4. As argued by Hoskins et al. (1985), this surface closely follows the dynamical tropopause over a wide range of latitudes, and so allows us to compare directly the potential for Rossby wave propagation at different latitudes.

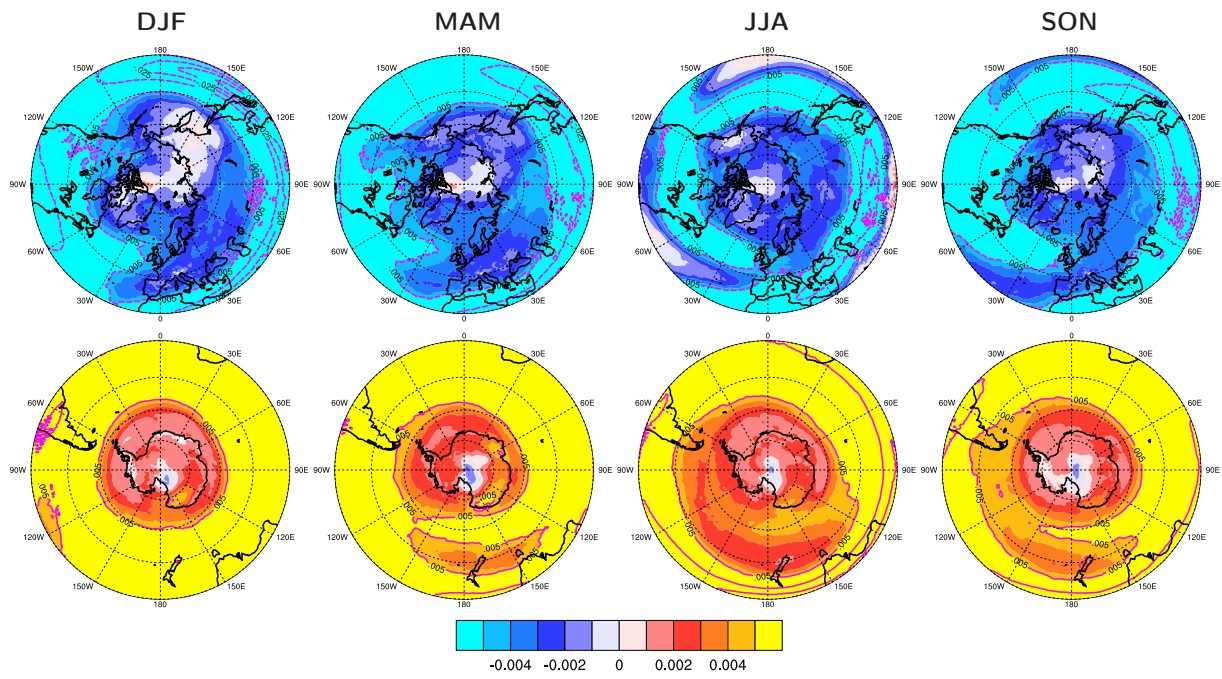

**Figure 4.** Seasonal climatologies of the meridional gradient of potential temperature $\theta$ on the 2PVU surface, from the ERA-Interim reanalysis. Contour lines are drawn every $0.005\,\mathrm{K\,km^{-1}}$, with negative contours dashed and shading used for smaller values.

$\theta_{PV2}$ gradients are strongest in the mid-latitude jet regions, as expected, and these features weaken and migrate polewards in summer in line with the waning of the subtropical jets. In winter the Arctic is far removed from the strong $\theta_{PV2}$ gradients

further south. The strongest Arctic winter gradients are seen over Alaska and the Nordic Seas, where the stationary ridges bring the relative vorticity gradients at the tail end of the storm tracks to high latitudes. Similar features are seen in the $K_s$ maps cited above. In these two regions, the $\theta_{PV2}$ gradients are around 20% of the strength of the gradients in the jet cores, indicating that conditions are much less favourable for Rossby wave propagation, but not prohibitive. At other longitudes, however, the gradients are considerably weaker. The weakness of these high-latitude tropopause gradients in winter is marked

in comparison to the stratosphere above, where strong relative vorticity gradients associated with the polar vortex are seen at subpolar latitudes.





As the circulation contracts towards the pole in summer and autumn, the $\theta_{PV2}$ gradients over the Arctic actually increase, despite the weakening of the jets. In these seasons the gradients at 70°N can be up to 50% of the strength of the midlatitude gradients, suggesting that polar Rossby wave dynamics may play a larger role in these seasons. The summer $\theta_{PV2}$ gradient in the Barents-Kara Sea region is stronger than the winter gradient across much of mid-latitude Europe, for example. Interestingly, Europe stands out in winter and spring for having a very weak $\theta_{PV2}$ gradient, consistent with the high occurrence of Rossby wave breaking and blocking there (Barriopedro et al., 2006).

In all seasons, the high Arctic poleward of 75°N has very weak gradients, indicating that Rossby waves are very unlikely to flourish there. At subpolar latitudes around 60-70°N, however, there seems to be potential for wave propagation, albeit weaker than in mid-latitudes. These $\theta_{PV2}$ gradients must rely on relative vorticity gradients associated with meridional shear of the zonal wind. As these winds are variable, this could be an explanation for the intermittency that has been suggested in some Arctic-midlatitude linkages (Overland and Wang, 2018; Siew et al., 2020). For example, the wavelike anomalies associated with the Siberian heatwaves of 2020 were observed to develop in a period of anomalously strong westerly flow across Siberia (Overland and Wang, 2021).

Southern hemisphere plots in Figure 4 show some non-negligible gradients over coastal regions of Antarctica, although these are weaker than those just discussed in the northern hemisphere. By comparison, the northern hemisphere gradients are likely associated with the temperature contrast between the Asian landmass and Arctic Ocean in summer, and the associated storm activity (Hoskins and Hodges, 2019). Despite the weak $\theta_{PV2}$ gradient, Patterson et al. (2020) showed that wave propagation around Antarctica can be enhanced by the topographic $\beta$ effect associated with the continental edge, an effect not captured in the tropopause diagnostic shown here.

## 4 Wave - turbulence interplay

Further insight into polar dynamics can be gained from considering the interplay of atmospheric turbulence with Rossby waves. Charney (1971) pioneered the theory of geostrophic turbulence as a descriptor of atmospheric circulation. The geostrophic relation is found to restrict the flow such that energy cascades up to larger spatial scales, as commonly seen in two-dimensional turbulence. Observations tend to support geostrophic turbulence as a relevant mechanism, particularly at synoptic and smaller scales (Boer and Shepherd, 1983). However, the cascade to larger scales is obstructed by the $\beta$-effect, as Rossby wave propagation becomes more important at larger scales. This is because the group velocity increases with the length scale of the perturbation (equation 6), so that disturbances can propagate away as Rossby waves rather than growing further (Rhines, 1975). The $\beta$-effect, or more generally the PV gradient, damps the growth of meridional length scales in particular, anisotropising the flow as a result and leading to the zonalisation which is characteristic of an atmosphere with jet streams (Vallis and Maltrud, 1993).

Waves and turbulence are thought to co-exist over a wide range of scales (Sukoriansky et al., 2007), and we hypothesise that the balance between these is different in polar regions than in midlatitudes. As the PV gradient is weaker in the polar regions, we expect Rossby waves to play a weaker role there and for turbulence to account for a greater share of atmospheric variability.



Since wave propagation is less efficient at high latitudes, this suggests that energy can continue to grow further to larger scales there, particularly in the meridional direction. The flow is hence expected to be closer to isotropic than it is at lower latitudes.

These predictions are now tested using a spectral analysis of reanalysis data. Figure 5 shows the vertically integrated eddy kinetic energy as a function of latitude and zonal wavenumber. Northern hemisphere data from all seasons are included, though similar results are obtained if the analysis is restricted to winter, for example. The time-invariant stationary waves have been

removed. This figure shows that energy is concentrated in mid-latitudes where baroclinic instability leads to strong eddy growth in the synoptic scales around wavenumbers 6-10. At these latitudes the spectrum flattens for wavenumbers lower than 6, an indication of the inhibition of the upscale energy cascade as described above (Boer and Shepherd, 1983).

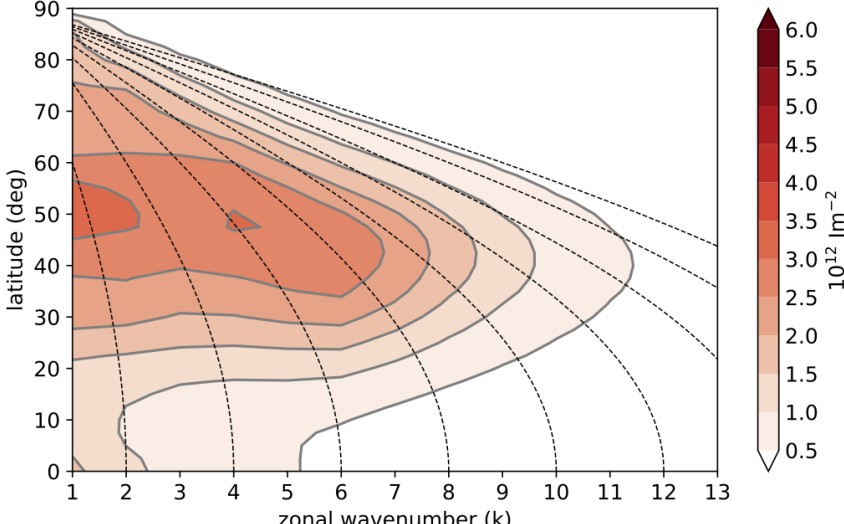

**Figure 5.** Vertically integrated eddy kinetic energy as a function of zonal wavenumber and latitude. Data is from the ERA-Interim reanalysis using all calendar months, and the time mean stationary waves have been removed. Dashed lines connect points with the same zonal length scale.

The polar latitudes in Figure 5 exhibit weaker eddy kinetic energy and the spectrum shows less indication of flattening at low wavenumbers. This suggests that the cascade to larger scales may indeed be less inhibited, as hypothesised, due to the

weaker PV gradient. The situation is unclear, however, because the Arctic is not an isolated region of the atmosphere but interacts frequently with the mid-latitudes through exchanges of air masses and weather systems (e.g. Sorteberg and Walsh, 2008; Kolstad et al., 2009). The dashed lines in Figure 5 connect points of constant physical length scale, so that an air mass passively advected across latitudes might be expected to follow one of these lines. The close association between these lines and the energy contours suggests that poleward energy transport from the mid-latitudes is likely important in shaping the polar

spectra.

A clearer indication is given by consideration of the flow isotropy. Pure geostrophic turbulence is isotropic in terms of zonal and meridional kinetic energy, while wave motion is often strongly anisotropic. Figure 6 shows the eddy anisotropy metric





$M = (u'^2 - v'^2)/2$ on the same axes as Figure 5. $M$ is the negative of the zonal component of the E vector of Hoskins et al. (1983), and specifically measures the anisotropy between the $x$ and $y$ directions.

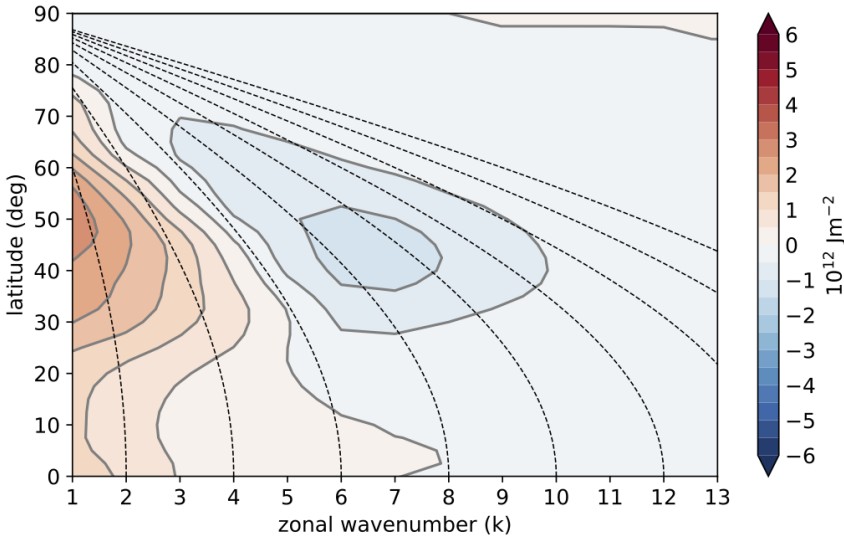

**Figure 6.** The eddy anisotropy metric $M = (u'^2 - v'^2)/2$, presented in the same format as Figure 5.

Mid-latitude flow at high wavenumbers exhibits negative values of $M$, indicating that the flow is dominated by meridional velocities. This is indicative of meridionally extended wave-fronts and hence zonally propagating waves. This is consistent with the eastward-pointed E vectors and downstream wave activity flux associated with high-frequency midlatitude eddies in Hoskins et al. (1983). Low-frequency eddies are instead zonally elongated with upstream-pointed E vectors, and these correspond to the positive values of $M$ at low zonal wavenumber. Moving to higher latitudes, however, we see that the magnitude

of $M$ decreases across the wavenumber spectrum. This shows that, as predicted, the circulation in the polar region is more isotropic than at low latitudes, with kinetic energy split more evenly between meridional and zonal wind. Similarly, the results of Barnes and Hartmann (2012) show that eddy length scales also become isotropic at high latitudes, while in mid-latitudes the zonal scale is on average larger than the meridional scale. This is consistent with the inhibition of the upscale cascade in the meridional direction especially.

Rossby waves and turbulence clearly do co-exist at most latitudes, but these results suggest that the polar region is characterised by weaker wave structures in favour of more isotropic vortices, as compared to the mid-latitudes. The presence of isotropic, often persistent polar vortices has been noted before (e.g. Cavallo and Hakim, 2010; Aizawa and Tanaka, 2016). Equally, persistent and slow westward-moving wavelike features have also been observed at relatively high latitudes (e.g. Branstator, 1987; Kushnir, 1987; Dunn-Sigouin and Shaw, 2015), consistent with the presence of some anisotropy in the 60-

70°N band in Figure 6. Indeed, such quasi-stationary waves in the subpolar zone may arise as a natural consequence of the cascade. Hendon and Hartmann (1985) note the similarity between the stationary wavenumber $K_s$ of equation 11 with the



Rhines scale postulated to relate to the inhibition of the upscale cascade ($\sqrt{\beta/2u_{rms}}$). This similarity depends on the root mean square wind speed $u_{rms}$ being similar to the mean speed, which seems reasonable. Hence, the upscale cascade is likely to deliver Rossby waves to the subpolar regions at length scales for which the waves will be close to stationary.

## 5   Correlation structures

We now show some selected examples of correlation analysis to reveal typical observed flow structures as a function of latitude. One-point correlation analysis has proved a useful method for extracting atmospheric flow structures in a range of applications (Blackmon et al., 1984). Following Chang and Yu (1999) and Wirth et al. (2018), we choose not to filter the data before analysis, as this can distort wave packets. As a consequence, the correlations shown here simply represent the most dominant structures out of a wide range of atmospheric variability. The meridional wind is often used in these analyses to highlight wavelike features. Here, instead, we use geopotential height to give equal weighting to zonal and meridional variations, and to avoid artificial signals at high latitudes where the meridians converge.

Figure 7 shows Hovmöller plots summarising the correlation analysis for a sample of latitudes in southern hemisphere summer (DJF), which was chosen as a relatively uncomplicated setting, with weak stationary waves and a uniform westerly jet in mid-latitudes. These were created by performing lag correlations of the 500hPa geopotential height along a latitude circle with the height at the base point, and this was repeated for base points spaced every 30° of longitude before averaging. Correlations were calculated for each DJF season over the period 1979-2018 using NCEP2 reanalysis data, and then averaged over the set of years.

The evolution at 45°S shows the classic signature of a dispersive mid-latitude Rossby wave packet, as documented by Chang and Orlanski (1993) for example. Individual centres of action proceed to the east over time following the phase speed of the wave. The amplitude of the wave packet evolves according to a group speed which is directed eastward and faster than the phase speed. Hence, new systems are formed downstream of, and at the expense of existing ones, in the process of downstream development. As described in section 2.2, this is in line with Rossby wave theory in which the eastward group velocity is greater than the phase velocity.

Further poleward, for base points at 60°S, a qualitatively similar evolution is seen. The wave packet structure is weaker, though, with only three centres of action seen on this latitude circle. On average, the flow here has some wavelike structure but the role of downstream development in the decay of the central feature is less clear. Moving further poleward again, to 75°S, the picture changes dramatically. Correlations are now positive over the whole of this domain, so that no wavelike structure is seen on average. These correlations are stronger in the vicinity of the central time and location but the weak correlations at longer lags and distances suggest a connectivity of the base point to all other points along the latitude circle. The convergence of meridians near the pole likely plays a role in this, and also spatial correlation maps (not shown) indicate that annular mode structures increasingly appear in these remote correlations as the base latitude is increased (cf Gerber and Thompson, 2017). The local correlations suggest the presence of a coherent vortex drifting upstream, towards the west, embedded within the annular mode.

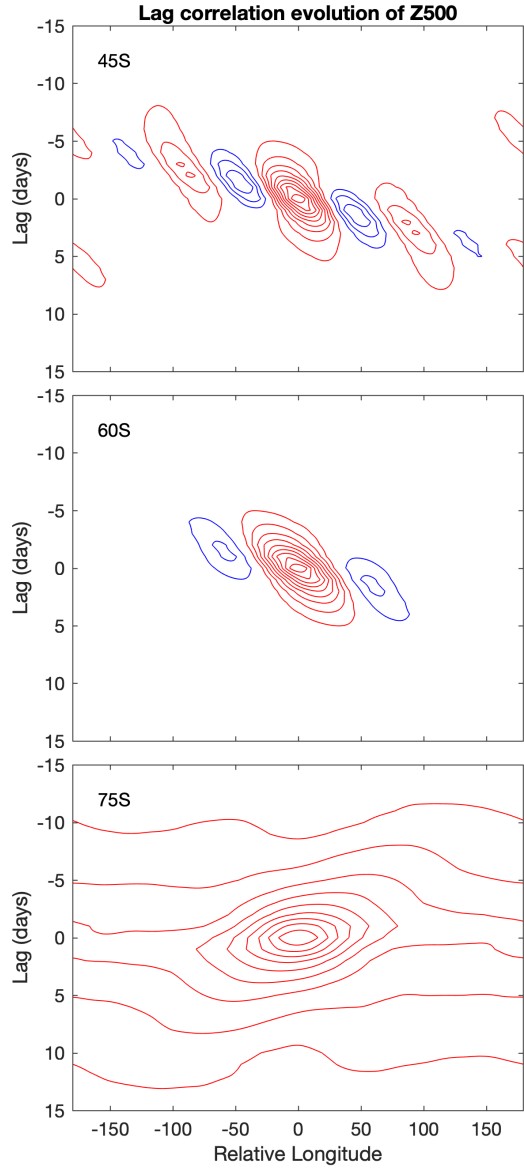

**Figure 7.** Examples of lag correlation analysis of daily mean Z500 from the NCEP2 reanalysis applied to the southern hemisphere in DJF. For each latitude (45, 60 and 75°S), analysis was performed for 12 base points equally spaced around the hemisphere and then the resulting correlation patterns averaged. The contour interval is 0.1 with the zero correlation lines omitted.

As an example of the spatial patterns associated with high latitude variability, Figure 8 shows a time sequence of correlation maps for base points at a latitude of 65°, this time in the northern hemisphere to demonstrate that wavelike features are also seen there. In this case, the central anomaly gradually moves upstream over time, to the west, as seen in the 75°S case above.

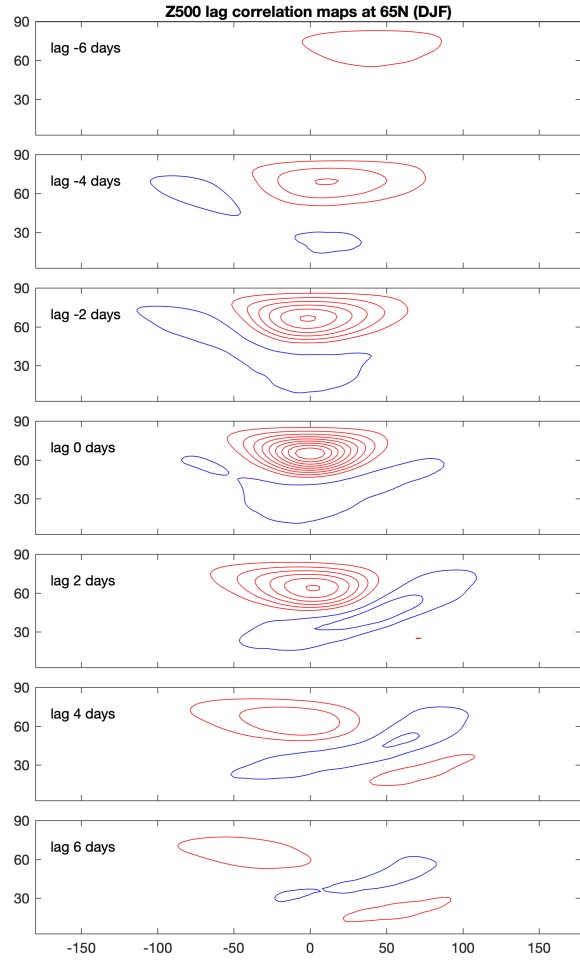

**Figure 8.** Lag correlation analysis of daily mean Z500 from the NCEP2 reanalysis. Shown is the average of 12 correlation maps calculated at $65°N$ for 12 base points equally spaced in longitude. The contour interval is 0.1 with the zero correlation lines omitted.

There are some indications of a wave packet passing through this feature, strengthening and then weakening it in turn, even if the correlations away from the central feature are relatively weak. The path of this wave arcs in from the southwest and then

departs toward the southeast, consistent with a poleward directed ray being turned at high latitude by the $\beta$-effect. The path of this wave has a strong meridional component, as opposed to the predominantly zonal propagation commonly seen at lower latitudes. As described in section 2.5, this seems consistent with the presence of a westward phase velocity which favours more meridional wave paths. There is also a weak negative anomaly to the south of the central feature, as the behaviour at this latitude starts to project onto the annular mode, and overall the pattern resembles the variability at intermediate timescales

over Greenland described by Rennert and Wallace (2009). Similar structures to those in Figure 8 are seen in both hemispheres,





albeit with some variation between locations. Annular mode signatures, for example, are most apparent in the Atlantic and Pacific storm track regions. A full analysis of the variety in these structures is left for future work.

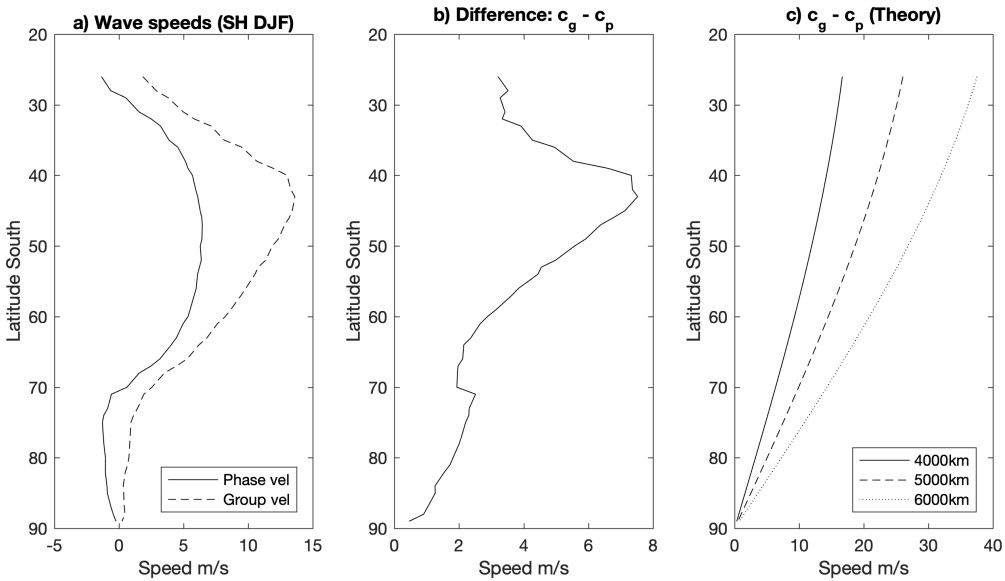

**Figure 9.** Phase and group velocities (a) and their difference (b) derived from the Z500 correlation analysis. Predicted differences using barotropic theory for some example zonal wavelengths are shown in (c).

To quantify some of the differences across latitudes, phase and group velocities were derived from the composite Hovmöllers and presented for the southern hemisphere in Figure 9a. These were calculated by finding the maximum correlation at lags of

up to $\pm 2$ days and performing a linear fit through these points. For the group velocity, the same approach was taken on the wave envelope, which was calculated using the Hilbert transform as in Ambaum (2008). Note that the semigeostrophic transform of Wolf and Wirth (2015) is not required, since this correlation analysis samples both cyclonic and anticyclonic conditions at the base point. Also, while we label the resulting speeds as phase and group velocities, the underlying features at very high latitudes do not appear predominantly wavelike.

The resulting profiles show several features which are apparent in the Hovmöller plots and also in qualitative agreement with the theory. Firstly, the group velocity is eastward and greater than the phase velocity at all latitudes. Both velocities are largest in the mid-latitudes and smaller at both low and high latitudes, as seen in other studies (Chang and Yu, 1999; Randel and Held, 1991) and consistent with the lower background wind speed $U$ there. At high latitudes the phase velocity becomes small and negative, as seen in the $65°N$ example in Figure 7. Negative phase speeds are consistent with equation 5 when $U$ and

$K$ are both small, and the prediction of Section 2.5 that westward propagation is likely to be common at high latitudes. The group velocity is small for latitudes above $65°$, as in the slowly retrograding but regionalised structures described by Branstator (1987).





One of the main predictions from section 2.2 was that waves would become less dispersive at higher latitudes, with a smaller difference between the phase and group velocities. This difference is shown explicitly in Figure 9b, confirming that in this analysis the two velocities are closer together at higher latitudes. This supports the qualitative impression from the Hovmöller plots that waves at higher latitude are less dispersive. Figure 9c provides an indication of the theoretical difference between the velocities, as given by equation 6 using example constant choices of wavelength at all latitudes. This bears some qualitative comparison to the observed difference, but note that an unrealistically short wavelength has to be used in order for the magnitude to be similar to that observed in Figure 9b. Additional discrepancies could arise from differences in the relative vorticity gradients, especially near the jets, and also because of variations in the typical Rossby wave scales with latitude (e.g. Barnes and Hartmann, 2011). This theory is clearly highly simplified and cannot quantitatively explain the observed speeds, but may still be qualitatively useful (cf Chang and Orlanski, 1993; Branstator and Held, 1995; Ambrizzi and Hoskins, 1997). It suggests that the less dispersive nature of high-latitude waves arises directly from the small value of $\beta$. Very similar behaviour can be seen in the results of Fragkoulidis and Wirth (2020), using a Fourier-based method to diagnose wave packets in reanalysis data. Their Figure S12 shows that the difference between zonal phase and group velocities decreases towards zero in both polar regions and in each season.

## 6   Rossby wave forcing at high latitudes

The previous sections have shown that Rossby waves can and do play a role in polar dynamics. A further question is whether Rossby waves can be forced, or triggered at high latitudes. It does appear so, given that several studies have noted Rossby wave propagation in response to localised Arctic perturbations in a range of models (e.g. Honda et al., 2009; Xie et al., 2020), often diagnosed using a metric of Rossby wave activity flux, such as that of Takaya and Nakamura (2001). The wave propagation is quite weak in such experiments, in line with the weaker PV gradients shown in Figure 4, but not negligible. In some cases, waves appearing to come from the Arctic may have originated elsewhere (Sato et al., 2014; Sorokina et al., 2016), and potentially been reinforced by heating in the Arctic (Gong et al., 2020). In this section we discuss possible mechanisms by which a stationary Rossby wave response might be triggered from the polar regions, for example by the diabatic heating associated with a local reduction of sea ice in the Barents-Kara region.

Conceptually, the circulation response to localised forcing can be divided into a direct effect, which is apparent within days of the forcing starting, and an indirect effect which develops on a timescale of weeks rather than days (Deser et al., 2004, 2007). The indirect effect often involves adjustment of the baroclinic eddy statistics and hence feedbacks onto the large-scale circulation. Such a storm track response is less likely at very high latitudes, but some studies have suggested it to play a role (Inoue et al., 2012). In response to a mid-latitude SST perturbation, Deser et al. (2004) observe a direct response which resembles the expected linear response of the atmosphere to mid-latitude heating. Linear theory has hence proved useful in understanding the response to heating at other latitudes, so here we consider the application of this theory to high latitude heating, such as might arise from a localised reduction in sea ice.





Hoskins and Karoly (1981) used a linear, steady state $\beta$-plane system to describe the linear response to heating in the tropics and the mid-latitudes, comprising vorticity and potential temperature equations given by

$$\bar{u}\zeta'_x + \beta v' = fw'_z, \tag{13}$$

$$\bar{u}\theta'_x + v'\bar{\theta}_y + w'\bar{\theta}_z = \frac{\theta_0}{g}Q, \tag{14}$$

where a prime denotes a deviation from the climatological mean, indicated by an overbar, subscripts denote partial derivatives and $Q$ is the anomalous diabatic heating rate.

In the tropics, heating is generally deep and is balanced most efficiently by ascent and the associated adiabatic cooling. The resulting deep vertical motion provides an efficient source of Rossby waves (Sardeshmukh and Hoskins, 1988). In mid-latitudes, horizontal temperature gradients are larger and heating is hence more efficiently balanced by horizontal advection. This is predominantly achieved through the establishment of a near-surface cyclone downstream, such that equatorward meridional wind brings relatively cool air over the forcing region to balance the heating.

Considering now the polar regions, ascent is strongly inhibited due to the high static stability, so balancing by vertical motion is unlikely. Assuming the required ascent could be realised, the resulting vortex stretching would then need to be balanced in the vorticity equation. In the tropics this is achieved through meridional flow and the associated $\beta$ term, but with a weak $\beta$ this is also difficult in the polar region. Alternatively, the vorticity equation could be balanced by the $\bar{u}\zeta'_x$ term through the formation of a vorticity anomaly upstream. However, the weak mean zonal wind $\bar{u}$ suggests this is also unlikely.

Instead, polar heating could be balanced by horizontal advection, as in the mid-latitude case. Anomalous equatorward wind would provide a cooling via the $v'\bar{\theta}_y$ term, however the meridional gradient $\bar{\theta}_y$ is weaker than in mid-latitudes. Balancing by the $\bar{u}\theta'_x$ term is also unlikely given the weak mean zonal flow.

Motivated by the structures of circulation response to sea ice loss in a range of numerical models presented in the literature, we suggest a different mechanism, namely that the direct response comprises a heat low. In this case the heating tendency is simply balanced by enhanced longwave emission, as also seen by Kim et al. (2021). Considering again a linear framework we can write this balance as

$$\gamma_b \frac{\partial T'}{\partial t} \approx -4\epsilon_a \sigma_B \bar{T}^3 T', \tag{15}$$

where the emissivity $\epsilon_a = 0.76$ and $\sigma_B = 5.67 \times 10^{-8}$ Wm$^{-2}$K$^{-4}$ (see, e.g. Barsugli and Battisti, 1998). Here we take $\gamma_b$ to be $10^6$ JK$^{-1}$m$^{-2}$, the heat capacity for a column of atmosphere in a one kilometre deep boundary layer, and the left hand side of the equation therefore represents an imposed heating in this layer. Hence, assuming a mean temperature $\bar{T} = 250$K, a heating rate of 1K day$^{-1}$ over this layer can be balanced by a warming of $T' \approx 4.3$K. These estimates are in good agreement with the simulations of Sellevold et al. (2016), who added heating to the high latitudes of a linear stationary wave model. This suggests that radiative heat loss can indeed be an efficient mechanism to balance heating at high latitudes.

Heat lows are characterised by a maximum in heating at or near the surface, as is commonly seen in numerical experiments of sea ice loss (Deser et al., 2010; Sellevold et al., 2016). The diabatic tendency in potential vorticity is positive below an imposed heating and negative above it (Hoskins et al., 1985). Hence, a heat low with a maximum in heating at the surface





exhibits a shallow surface cyclonic anomaly, consistent with the positive anomaly in potential temperature there, but then a deep anticyclonic anomaly above (Spengler and Smith, 2008). This corresponds to the classical monsoon structure of low

pressure at the surface and anticyclonic flow aloft. This heat low signature is seen in several numerical experiments of sea ice loss (Honda et al., 2009), including importantly the linear models used by Sellevold et al. (2016) and Deser et al. (2007), and in the direct/transient responses of Deser et al. (2004).

As further evidence of the importance of a heat low, we now present the results of a perturbation experiment using an idealised general circulation model. Following the experiments of Hell et al. (2020), we use a gray-radiation aquaplanet model

with a slab ocean and impose high-latitude heating via a prescribed convergence of oceanic $Q$ fluxes. A heating of $200\,\mathrm{Wm^{-2}}$ is applied to a patch spanning 90 degrees of longitude between $70°\mathrm{N}$ and $85°\mathrm{N}$. The experiment is allowed to equilibrate for 180 days after the Q-flux forcing is turned on, after which daily output is saved for 720 days.

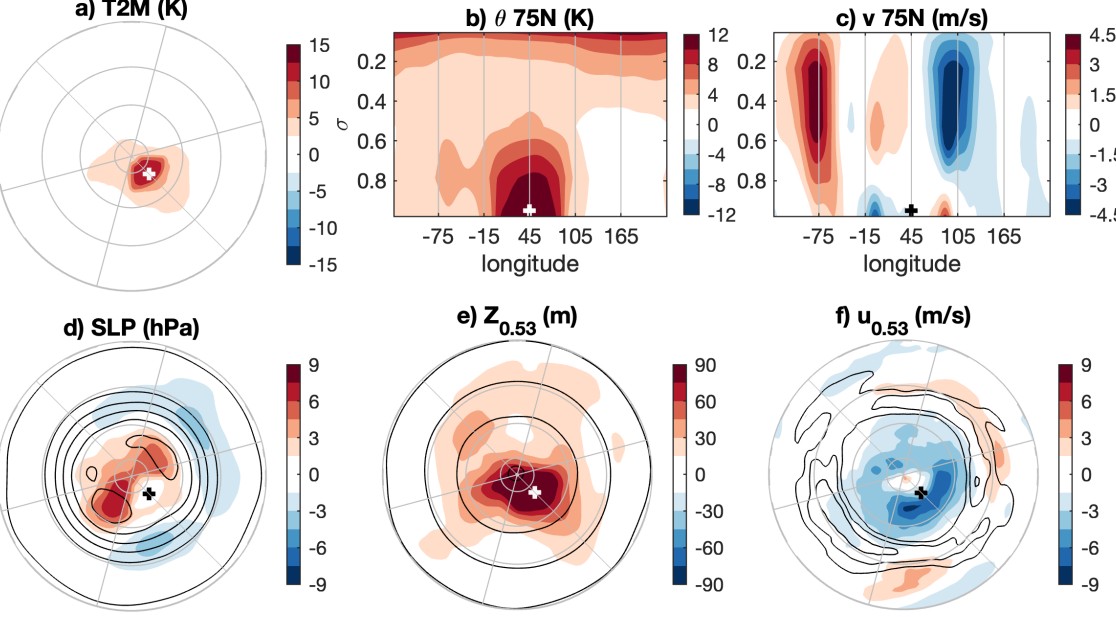

**Figure 10.** The response of an idealised GCM to localised heating at high latitudes. Contour lines in panels d-f indicate the mean state, with contour intervals of d) 5 hPa, e) 200 m, f) $1\,\mathrm{m\,s^{-1}}$. A cross marks the location of the imposed heating.

The response to this heating is summarised in Figure 10. A strong warming anomaly is co-located with the heating (10a) and decays with height (10b) so that most of the warming is seen below $600\mathrm{hPa}$ (though note the stratospheric warming

associated with a weakened polar vortex as in Hell et al. (2020)). A vertical section of meridional velocity through the heating (10c) shows the clear signature of the heat low, comprising a cyclonic anomaly around the heating location at the surface but an anticyclonic anomaly spanning the depth of the troposphere above. The depth of this anomaly increases the likelihood





of successfully triggering a Rossby wave response. There are other structures in the response though these are not clearly
wavelike; in this simulation the zonal jet is located at a lower latitude (10f) so the relative vorticity gradient is likely not strong
enough to support this. In terms of pressure fields, the response is a local minimum in sea level pressure coincident with the
heating (10d) with a local maximum in mid-tropospheric height above (10e). More generally, the response projects onto the
negative phase of the Northern Annular Mode (10d, e, f), as often seen in response to strong polar heating such as this.

To summarise, the linear mechanisms which often balance an imposed heating at other latitudes are less efficient at polar
latitudes. Instead, we suggest that the direct response to high-latitude near-surface heating is a simple radiative damping which
sets up a heat low structure, as shown in Figure 10. With a heating maximum near the surface, the heating decays with height
in the troposphere, hence forcing an anticyclonic circulation. This diabatically forced anticyclone comprises a PV anomaly
which has the potential to develop into a Rossby wave response, as seen here and in some numerical model experiments in the
literature.

However, this wave response is typically very weak in models, because of the weakness of the high-latitude PV gradient,
and may be sensitive to the depth of the warming (He et al., 2020). The particularly high stability in the polar winter suggests
that a heating deep enough to trigger a Rossby wave is more likely to occur in other seasons such as summer and autumn.
The magnitude of the response will depend on the strength and depth of the relative vorticity gradients, though in any case the
ray tracing discussion of Section 2.5 suggests that wave amplitude is likely to weaken further as the wave propagates to lower
latitudes. Finally, the structure of the heat low may also lead to a weak negative feedback on the surface heat flux anomaly
which created it, here supposed to be due to an area of anomalously low sea ice cover. By trapping the atmospheric heat
anomaly near the surface, this structure will slightly offset the temperature contrast between the cold atmosphere and warm
ocean, hence leading to a negative feedback on the anomalous upwards surface heat fluxes (Hendon and Hartmann, 1982).

## 7 Concluding remarks

Much of extratropical large-scale variability can be interpreted as arising from the interplay of Rossby waves and geostrophic
turbulence. Energy cascades to larger scales in geostrophic turbulence, but the meridional length scales in particular are re-
stricted by the beta effect and associated efficient Rossby wave propagation. Beta is small in the polar regions and hence the
flow there is relatively more vortex-dominated than the wave-dominated flow at lower latitudes. This is seen in the morphing of
correlation structures from wavetrains into monopoles as the latitude is increased, and the correspondingly more isotropic eddy
structures. The flow more often resembles the advection of individual vortices than the propagation of waves along vorticity
gradients.

Despite this, some wavelike features are still apparent at relatively high latitudes of up to around $75°$, and these offer
some insight into persistent polar/subpolar flow patterns such as those which contributed to the 2020 Siberian heatwaves. For
example, the slow westward progression of circulation anomalies is consistent with wavelike propagation, or 'beta-drift', and
cannot be explained as advection by the background flow which is generally westerly, especially at upper levels. Although
beta is weak in the polar regions, relative vorticity gradients can enhance the potential for wave propagation. Hence, potential



vorticity gradients at these latitudes are small but not negligible, particularly in summer and autumn. As a result, turbulent and wavelike flows co-exist to some extent, even at these high latitudes, just with a different balance than in mid-latitudes. Importantly, however, while these two regions have different characteristics they are not isolated systems, instead engaging in frequent exchanges of air masses and weather systems with each other.

Rossby waves are readily triggered by deep convection in the tropics and by baroclinic growth in the mid-latitudes, and these waves often propagate to higher latitudes. Waves are typically turned in subpolar latitudes and refracted back towards the stronger mid-latitude PV gradients, and then often further equatorwards. Only the longer waves reach the highest latitudes and the convergence of wave activity density can lead to an increase in wave amplitude there. Typical length scales are such that the subpolar anomalies are often quasi-stationary or exhibit a slow westward phase velocity under the action of the beta

effect, as in the 2020 example in Figure 1. Waves can also break in these regions, as the zonal wind is often weak and there is a convergence of wave activity, leading to irreversible overturning of potential vorticity contours and isolated anticyclones at high latitudes.

    It is worth highlighting that these processes can naturally lead to instances of amplified mid-latitude waves co-existing with persistent polar circulation anomalies, without there necessarily being any causal influence from the Arctic (see also Kelleher

and Screen, 2018). This may be relevant to the discussion of the mid-latitude response to Arctic warming, but has not been investigated further here. There is also potential for Rossby waves to be triggered by localised heating at high latitudes, via a local heat low response, although this mechanism appears considerably weaker than those which trigger waves at lower latitudes.

    One of the more practical conclusions of this paper is that the behaviour of Rossby waves at high latitudes can contribute

to the longer timescales evident there, as demonstrated in Figure 2. At mid and low latitudes, Rossby wave dispersion is an efficient mechanism which contributes to the decay of eddies, by fluxing wave activity downstream. However, high latitude waves are less dispersive, so that downstream development does not play a strong role in the decay of eddies. The lack of this decay mechanism is suggested as a key reason for increased flow persistence at high latitudes.

    Persistent polar cyclones are a clear example of this. Some cyclones are generated within the Arctic region but many are ad-

vected into it from lower latitudes, leading Serreze and Barrett (2008) to describe the Arctic as a 'collection zone' for cyclones. In either case, Arctic cyclones can have considerably longer lifetimes than those at lower latitudes (Hoskins and Hodges, 2002), in line with the persistence statistics shown in Figure 2. Some of the notably persistent cyclones have been linked to the frequent merging of existing vortices with newer ones (Yamagami et al., 2017). It is tempting to identify this process with that of vortex merging in the upscale energy cascade of geostrophic turbulence, though this may be overly simplistic as three-dimensional

vortex structures could be important. In any case, the longevity of these cyclones would clearly be significantly reduced if Rossby wave dispersion were active as a means of eddy decay, as it is in mid-latitude cyclones.

    The annular modes provide another example of the importance of the polar dynamics discussed here. These emerge as the leading modes of variability in extratropical flow in observations and models, and are characterised by zonally symmetric anomalies in wind, height and surface pressure. Gerber and Thompson (2017) showed that this annularity can emerge from

dynamical relationships leading to positive correlations along latitude circles, but also from statistical relationships. Even when





the mid-latitude flow exhibits wavelike, rather than annular correlation structures, the coherence of flow in the polar region helps the annular mode structure to emerge from empirical orthogonal function (EOF) analysis. In some cases, exclusion of the polar region from the domain leads to EOF patterns dominated by mid-latitude waves rather than meridional dipoles (Spensberger et al., 2020). We suggest that the emergence of annular EOF patterns is favoured by the prevalence of isolated vortices in the

polar regions, which lead to same-signed circulation anomalies in contrast to the alternating signed anomalies characteristic of wavelike flow.

It is hoped that this paper motivates further work on the large-scale dynamics of the polar regions and its role in weather and climate. For example, the brief analysis presented here reveals little of the variation and the regional detail of polar flow structures. One limitation of the correlation analysis used here is its symmetrical nature, combining both cyclonic and

anticyclonic anomalies into the same structure. We anticipate that polar flow features might exhibit considerable asymmetries, and there is some evidence of this in skewness statistics (Luxford and Woollings, 2012), extreme event distributions (Messori et al., 2018; Papritz, 2020) and diabatic effects (Cavallo and Hakim, 2010). After all, while it is possible to create both warm anomalies and anticyclonic circulations simply by moving an air mass from lower latitudes into the polar region, there is not an equivalent mechanism for generating cold anomalies or cyclones there.

## 8   Appendix

The stirred barotropic model used in this study is identical to that introduced by Vallis et al. (2004) and subsequently employed by Barnes and Hartmann (2011) and Kidston and Vallis (2012) amongst others. The model is forced by random stirring which is imposed in the mid-latitudes to mimic the generation of eddies through baroclinic instability. These eddies then act as a vorticity source to maintain a mid-latitude jet.

The non-divergent barotropic vorticity equation on the sphere is given by

$$\frac{\partial \zeta}{\partial t} + \frac{u}{a\cos\phi}\frac{\partial \zeta}{\partial \lambda} + \frac{v}{a}\frac{\partial \zeta}{\partial \phi} + \beta v = S - r\zeta + \kappa \nabla^4 \zeta, \tag{16}$$

in which $\zeta$ is the relative vorticity, $a$ is the radius of the Earth, and $\lambda$ and $\phi$ are the zonal and meridional co-ordinates respectively. The random stirring, $S$, is defined following Vallis et al. (2004) by an Ornstein-Uhlenbeck stochastic process. For each timestep, $i$, a random number, $Q^i$, is drawn from a uniform distribution in the range $[-7 \times 10^{-11}, 7 \times 10^{-11}]$. The stirring

is then calculated in wavenumber space via the following equation

$$S_{lm} = (1 - e^{2dt/\tau})^{1/2}Q^i + e^{-dt/\tau}S_{lm}^{i-1}, \tag{17}$$

where $l$ and $m$ are respectively, the total and zonal wavenumber, $dt$ is the timestep (900 seconds), and $\tau$ is a decorrelation timescale equal to 2 days. The model is stirred with total wavenumbers between 4 and 12 and zonal wavenumbers less than 4 are excluded. The stirring is then transformed from spectral space into real space and multiplied by Gaussian functions of half



width 12°, centred on 45° in each hemisphere, such that the stirring is only applied in the mid-latitudes to represent the storm tracks. Linear Ekman damping is employed with coefficient, $r = \frac{1}{5} day^{-1}$ and the loss of vorticity at the smallest, unresolved scales is simulated via 4th order hyperdiffusion. The model is integrated at T42 resolution for 10,000 days.

*Author contributions.* TW conceived and led the study and the writing of the paper, in collaboration with CL. The observational data analysis was performed by TW, MD, ED-S, KE, CM and MP. Idealised model experiments were performed and analysed by MP, MH and CL. All

authors contributed to the writing of the paper.

*Competing interests.* At least one of the authors is an editor/co-editor of Weather and Climate Dynamics.

*Acknowledgements.* MD, CM and TW were supported by NERC grants NE/N01815X/1, NE/S004645/1 and NE/M005887/1, and CL and TW by Research Council of Norway grants 255027, 276730 and 310391.



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
