# Peer review of "The role of Rossby waves in polar weather and climate"

_Weather and Climate Dynamics, 2022_

## Author Response (AR1)

We would like to thank the reviewers for their supportive comments. Their suggestions have been very useful for improving the paper, and we note below how we have addressed these.

**Reviewer 1**

This is an important contribution that notes specific theory and conclusions regarding Arctic atmospheric dynamics

Some are more obvious like weakening Rossby waves, and some are more interesting like the preference for very low wave numbers and importance of local long wave radiation.

1. The title and conclusions are about the Arctic. Yet some of the motivation is from Arctic-midlatitude linkages that are not really discussed

I would cut back on this connection.

Yes, we have cut back on this connection as suggested. We removed the more general text from the second paragraph of the introduction on this, and just retained the specific list of motivation points that are more relevant. See lines 30-35 in the differenced PDF.

2. There is five pages of background theory. This could be more streamlined removing secondary comments. Move some more to Appendix? Focus on what is important for conclusions. The manuscript needs a better flow and shortening.

Yes, we have moved the section on beta plane limitations to an Appendix (previously section 2.3). This was very much a side issue and so this has helped to strengthen the flow of the paper as well as streamline the main body of the text.

3. Paper is well written. Anticipated conclusions are mentioned at the beginning of each section before going into details.

Thank you for the comment. We hope that the streamlining you have suggested has helped to improve the accessibility further.

4 Not sure all of these authors made substantial contributions, but leave this to main author and editor.

Well, many hands make light work! The authors have all made distinctive contributions as outlined in the Author Contributions section.

**Reviewer 2**

The authors discuss theoretical insights into the creation and propagation of Rossby waves at high latitudes. The paper provides a much needed foundation for an active

area of research, is well written and a substantial contribution. I recommend the manuscript for publication with only a few minor comments:

1. L 26/27 The authors later refer to recent research discussing to what extent 'annular mode' variability is caused by a physically symmetric phenomenon or emerges from EOF analysis despite the actual variability being more regional. It would be good to present this consistently throughout the paper.

   Good suggestion, we have adjusted the description in the section of correlation structures to reflect this, and to signpost the annular mode discussion coming up later in the conclusions section. We also added a note to the end of the introduction on this. See lines 23-24, 99 and 388-390 in the differenced PDF.

2. L 53/54 Having read the paper, my impression now is that Rossby waves do play a role in the Arctic, but might be a little less important at polar latitudes then suggested in some of the above-cited literature. With this in mind, the sentence might be read as a subtle criticism suggesting precisely this. If this is intended, the criticism might be a bit too subtle at first reading. Otherwise, the authors might want to elaborate in the discussion section (or explicitly open the question) to what extent the suggested mechanisms for Rossby waves to play a (key) role in Arctic climate in the literature appear plausible in the light of their analysis.

   No, this wasn't intended as a criticism, just as a motivation. We have reworded this to avoid that impression. In line with Reviewer 1's comments we have also reduced the reference to polar-midlatitude linkages in the introduction. In the conclusions section we have expanded the note on Arctic-midlatitude linkages to make clear that our specific contribution is on the mechanism for triggering Rossby waves from high latitudes. See lines 30-35, 57-58 and 550-557 in the differenced PDF.

Section 6: In addition to the heat low mechanism, it would be interesting to refer to the formation of cold-core anticyclones through radiative cooling (Curry 1987)

Yes, that is an interesting and useful reference. The circulation response to their imposed cooling is similar in structure and magnitude (just opposite in sign) to our heating experiments, and we have added a note on this in section 6.

**Reviewer 3**

Summary

This paper presents a theoretical discussion, with illustrative examples, of the role of Rossby waves in polar meteorology. It is a timely piece of research, since there have been a large number of studies aiming to infer the dynamic response of the Atmosphere to polar amplification as well as increased focus on providing skillful weather forecasts for the polar regions, where skill has typically been lower than in

mid-latitudes. However, relatively few studies have attempted to put the large-scale atmospheric dynamics of the polar regions on a more theoretical footing.

I really enjoyed and learned a lot from reading this paper, however I think there are areas where it could be improved prior to publication, and I have made some specific suggestions below. My one concern with the paper was a tendency in the paper for arguments and analyses to ignore the seasonality which is large in polar areas leading to dramatic differences in large scale atmospheric structure and phenomena. This is touched on in the description of Figure 4, but I belive the manuscript would be improved by exporting this discussion of the implications to other sections.

Regarding seasonality, the environmental conditions in the polar regions are very different in DJF than JJA. For example although the Arctic atmosphere is stably stratified in winter almost 100% of the time, in the summer particularly over land it is only stably stratified about 50% of the time and inversions are much weaker when they occur (Serreze et al., 1992; Serreze and Barry, 2009). It made me wonder whether the potential for convection to trigger a Rossby wave response over continental regions of the Arctic in summer had been overlooked.

We also see very different dynamic behaviour in different seasons. i.e. very long-lasting large-scale tropospheric vortices in the summer, but smaller, shorter more intense vortices in winter (Vessey et al., 2022). Given that these differences exist I think the discussion should been revised to be more careful to mention to which season specific arguments apply to and not to jump between seasons when comparing figures. I will point out where I think this is an as issue in the text below.

We thank the reviewer for their very helpful suggestions, in particular on the seasonality aspects which we have found very useful. We have added in references and notes on the seasonality following these suggestions, at the locations outlined below.

Specific comments edits:

L2: involves to involve

Thanks – corrected.

L70: I would suggest removing the slightly tangential remark about the sea ice as the information content in the paper is already quite high.

Fine, this has been removed.

L80: If I understand correctly the difference in theta gradients shown in Figure 4 around the 65-75 band between DJF and JJA suggest quite different potential for Rossby wave propagation. It makes me wonder if interpreting Fig 2 with FIG 1 is meaningful. I suggest including the e-folding timescales for JJA in Fig 2 as well.

Figures 1 and 2 were just meant as illustrative examples rather than being quantitatively linked. We have added a note to the text to clarify this. To investigate further we should probably perform a more regional version of the e-folding analysis

to isolate the region of the Arctic frontal jet. We thank the reviewer for the good suggestion, but we will have to leave that for future work.

L85: change "in timescale in the observations" to "in the e-folding timescale of the observations"

Fine, this has been changed.

L273: This increase in temperature gradients coincides with the appearance of the Arctic frontal jet (Serreze et al., 2001; Crawford and Serreze, 2014; Day Jonathan J. and Hodges Kevin I., 2018) which I think is important to mention here. Recent studies have discussed the co-occurrence of quasi stationary waves and what they call "double jet" occurrence in high latitudes (e.g. Kornhuber et al., 2017). There is a bit of a disconnect in the literature produced by these different groups and it might be a good opportunity say something about this as well.

Yes, this is a very good suggestion. We have added some text here, including references as suggested (though we refer to Rousi et al (2022) which we think is more specifically relevant than Kornhuber et al). We have also added a note on the ongoing strengthening trend of the Arctic frontal jet, referring to Day and Hodges (2018) as well as Rousi et al. See lines 281-291 in the differenced PDF.

L337: Vessey et al (2022) quantify this in a systematic way over many cases showing by selecting the 100 most intense Arctic and N Atlantic storms and find the mean lifetime for these is 5.4 days for NA-DJF, 6.1 days for Arctic-DJF and 9.7 days Arctic-JJA.

Thanks for this useful new reference which we were not aware of. We have added it to the paragraph discussing cyclones in the Conclusions.

Section 6: It' probably important to mention that this section is really describing the situation in winter. In summer, from a thermodynamic perspective the position of the sea ice has much less of an influence on the turbulent exchange. Also the arguments related to static stability in the paragraph starting on L446 are most relevant for winter as already mentioned.

Yes, we have adjusted this to discuss the seasonality as suggested. We note that the forcing from sea ice is most relevant in winter and autumn, but also that this theory would apply to heating over continental high latitude regions in summer, as noted by this reviewer above. We have included the Serreze et al (1992) reference here. See lines 442-444 and 464-467 in the differenced PDF.

Crawford, A. D. and Serreze, M. C.: A New Look at the Summer Arctic Frontal Zone, J. Climate, 28, 737–754, https://doi.org/10.1175/JCLI-D-14-00447.1, 2014.

Kornhuber, K., Petoukhov, V., Petri, S., Rahmstorf, S., and Coumou, D.: Evidence for wave resonance as a key mechanism for generating high-amplitude quasi-stationary waves in boreal summer, Clim Dyn, 49, 1961–1979, https://doi.org/10.1007/s00382-016-3399-6, 2017.

Serreze, M. C. and Barry, R. G.: The Arctic Climate System, Cambridge University Press, 2009.

Serreze, M. C., Schnell, R. C., and Kahl, J. D.: Low-Level Temperature Inversions of the Eurasian Arctic and Comparisons with Soviet Drifting Station Data, J. Climate, 5, 615–629, https://doi.org/10.1175/1520-0442(1992)005<0615:LLTIOT>2.0.CO;2, 1992.

Serreze, M. C., Lynch, A. H., and Clark, M. P.: The Arctic Frontal Zone as Seen in the NCEP–NCAR Reanalysis, J. Climate, 14, 1550–1567, https://doi.org/10.1175/1520-0442(2001)014<1550:TAFZAS>2.0.CO;2, 2001.

Vessey, A. F., Hodges, K. I., Shaffrey, L. C., and Day, J. J.: The composite development and structure of intense synoptic-scale Arctic cyclones, Weather and Climate Dynamics, 3, 1097–1112, https://doi.org/10.5194/wcd-3-1097-2022, 2022.

**Community comment from Adam Scaife**

A small comment to request clarification of upstream propagation.
It is suggested (line 122-128) that long waves have the potential for westward propagation relative to the ground. However, I think this is potentially misleading as the simple theory discussed here does not allow westward propagation if the waves are stationary. This is easily shown by deriving an expression for Cgx and using the condition ω = 0 as in Scaife et al, QJRMS, 2017.
Please could the discussion be clarified here to make it clear that, *irrespective of the values of k and l*, it is only *transient* waves that have any possibility for westward propagation relative to the ground? If this is not correct I would equally interested to know!
Many thanks

Yes, we agree that for stationary waves the zonal group velocity has to be directed eastward, so long as k>0. Much of our discussion includes transient waves as well as stationary, and we have added a clarification on this distinction as suggested. See lines 134-136 in the differenced PDF.

---

## Author Response (AR2)

Referee#3 Report #2 "Line 258: Note that the Day and Hodges (2018) paper does not actually show an increasing trend in U250 in the Arctic frontal Jet region over the period of the reanalysis, rather a high correlation between inter-annual JJA U250 and dT/dy in the region. So this sentence needs a slight revision. For info: climate models do show an increase in U250 under e.g. RCP8.5 (see references in the paper)."

Thanks to the reviewer for spotting this error. Indeed, Day and Hodges demonstrate a trend in the temperature gradient but not in the zonal winds. We have adjusted the sentence to fix this, as suggested. We have also taken the opportunity to add a reference to a paper showing a stationary wave analysis of this region, that we should probably have been aware of.

"Climatological stationary wavenumbers of 6-8 are seen along the Arctic coast, with some potential for waveguiding there (Hoskins and Woollings, 2015, Figure 2). The Arctic land-sea temperature contrast has strengthened over the reanalysis period (Day and Hodges, 2018) and under continued anthropogenic warming this is expected to lead to a strengthening of the Arctic frontal jet, further increasing the potential for wave propagation. These changes have recently been implicated in the occurrence of persistent Rossby waves and associated heatwave events over Europe (Rousi et al., 2022)."